# Retrospecting the Antioxidant Activity of Japanese Matcha Green Tea–Lack of Enthusiasm?

**Iyyakkannu Sivanesan** [1] , **Judy Gopal** [2] , **Manikandan Muthu** [2] , **Sechul Chun** [3] **and Jae-Wook Oh** [4,*]

1   Department of Bioresources and Food Science, Konkuk University, Seoul 143-701, Korea;
    isivanesan@gmail.com
2   Laboratory of Neo Natural Farming, Chunnampet, Tamil Nadu 603 401, India; jejudy777@gmail.com (J.G.);
    bhagatmani@gmail.com (M.M.)
3   Department of Environmental Health Science, Konkuk University, Seoul 143-701, Korea;
    scchun@konkuk.ac.kr
4   Department of Stem Cell and Regenerative Biotechnology, Konkuk University, Seoul 143-701, Korea
*   Correspondence: ohjw@konkuk.ac.kr; Tel.: +82-2-2049-6271; Fax: +82-2-455-1044

**Abstract:** Matcha tea is a traditional Japanese tea that is said to possess ten times higher bioactive components and polyphenols than that of conventional green teas. Matcha is remotely popular among the global community and meagerly researched and infamous among the scientific population. It is the powdered form of green tea leaves that are directly suspended in hot water and drunk without filtration. Matcha is said to be one of the richest antioxidant sources naturally available. This review summarizes the available research publications related to matcha and compares the research accomplishments of green tea and matcha researchers. The fact that green tea is backed up by 35,000 publications while matcha has merely 54 publications to its credit is highlighted in this review for the first time. The future of matcha for tapping its enormous antioxidant activity and health potentials remains connected to the volume of scientific awareness and enhanced research attention in this area. If green tea has so much to offer towards human health and welfare, there is certainly room for more benefits from matcha, which is yet to be disclosed. As public awareness cannot be won without scientific approval, this review seeks that this gap may be bridged using essential knowledge gained from matcha applications and allied research.

**Keywords:** polyphenols; antioxidant activity; green tea; matcha tea; catechins; health benefits

## 1. Introduction

Tea is one among the three predominant beverages in the world. From time immemorial, it has been in use as a health product, stimulant, and medicine for the prevention of various diseases. Antioxidant, bacteriostatic, anti-cancer, anti-obesity, anti-diabetic, anti-cardiovascular, anti-infectious, anti-neurodegenerative effects, and regulation of lipid metabolism are few outstanding health benefits of tea [1,2]. Green tea is an offshoot of tea that has far outweighed the health benefits of teas in leaps and bounds. Figure 1 displays the benchmarks and milestones and accomplishments that have ensued from green tea research. The health-promoting activity of green tea is due to its polyphenol content [3], specifically from that of the flavanols and flavonols, which are about 30% of the dry weight of the fresh leaves [4]. Green tea contains four main catechins, i.e., (−)-epicatechin-3-gallate (ECG), (−)-epicatechin (EC), (−)-epigallocatechin (EGC), and (−)-epigallocatechin-3-gallate (EGCG). The catechin EGCG is the most active and abundant in green tea [5,6]. The health benefits of green tea are primarily attributed to (−)-epigallocatechin-3-gallate (EGCG) form of catechin [7,8]. It is published that a single cup of green tea brewed from 2.5 g of green tea leaves contains 240–320 mg of catechins; EGCG accounts for 60–65% of the total catechins in a cup of tea [9]. The theory of using this predominant catechin rather than the green tea extract has been addressed, and the inference that tea extracts are more stable than pure

epigallocatechin gallate has been established. This is owing to the fact that green tea extract contains various other antioxidant constituents that act as enhancers and stabilizers [10,11].

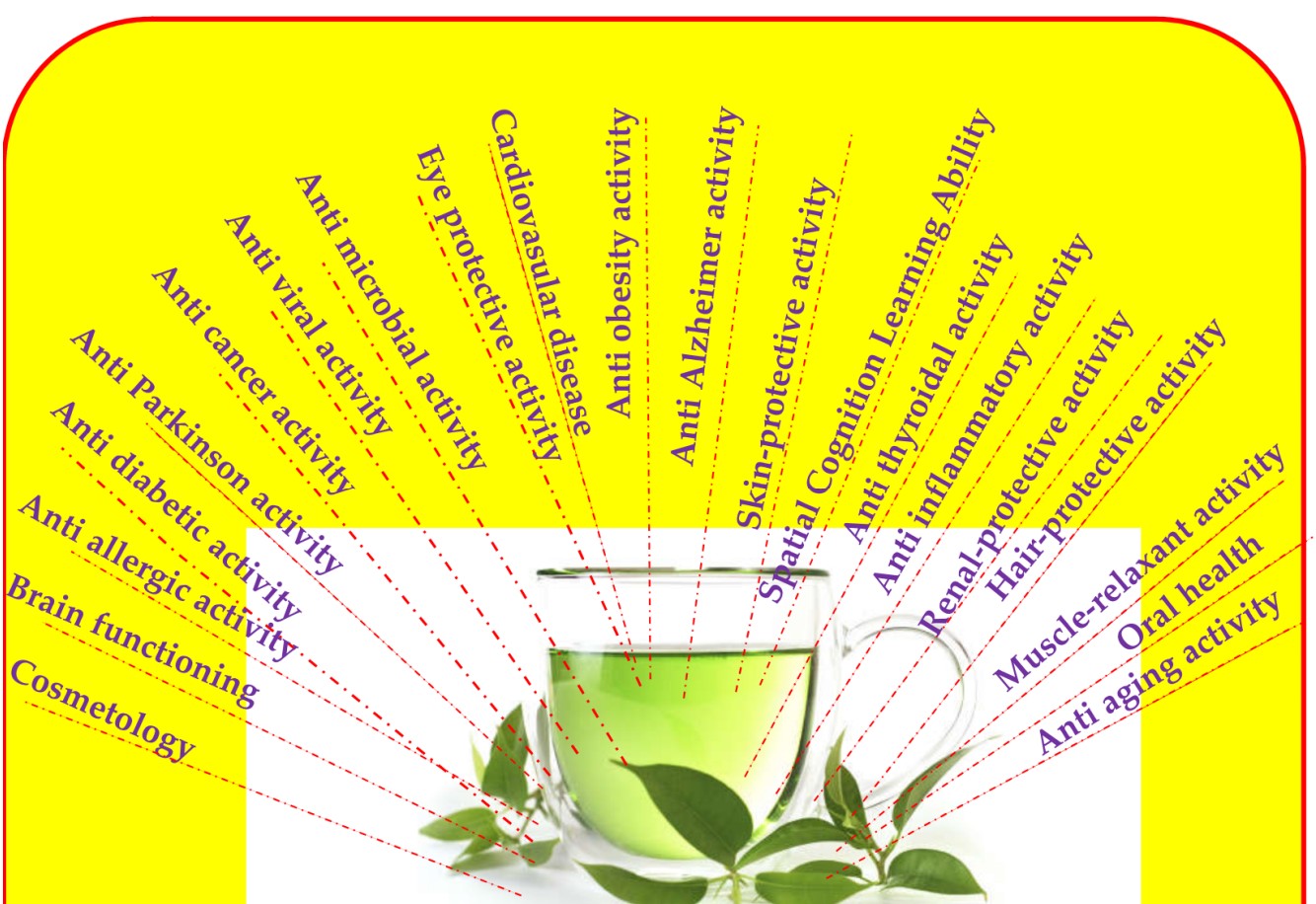

**Figure 1.** Overview of the health benefits of green tea.

Tea is divided into four categories based on their respective fermentation levels (Figure 2): white tea, green tea, oolong tea, and black tea [12]. White tea is obtained from young tea leaves and buds that are dried. Green tea is from mature leaves that are dried. Oolong tea is obtained from partially fermented mature tea leaves and black tea from fully fermented leaves [13,14]. Matcha is actually a variant of green tea differing in its growth condition and its post-harvest preparation. It is Japanese in origin [15–20]. The origin of both matcha and regular green tea is from *Camellia sinensis*. However, matcha requires different growth conditions than that of green tea. In case of matcha tea, the tea plants are grown protected from sunlight for about 20–30 days prior to harvest. The shade triggers a spike in chlorophyll levels, rendering the leaves dark green and possessing increased amino acids. Post-harvesting, the stem and veins of the leaves are removed and stone-ground into a fine, bright green powder known as matcha. The green tea is prepared using this powdered green tea, without straining [21]. Because the whole leaf powder is ingested, matcha is even higher in some substances, such as caffeine and antioxidants, than green tea.

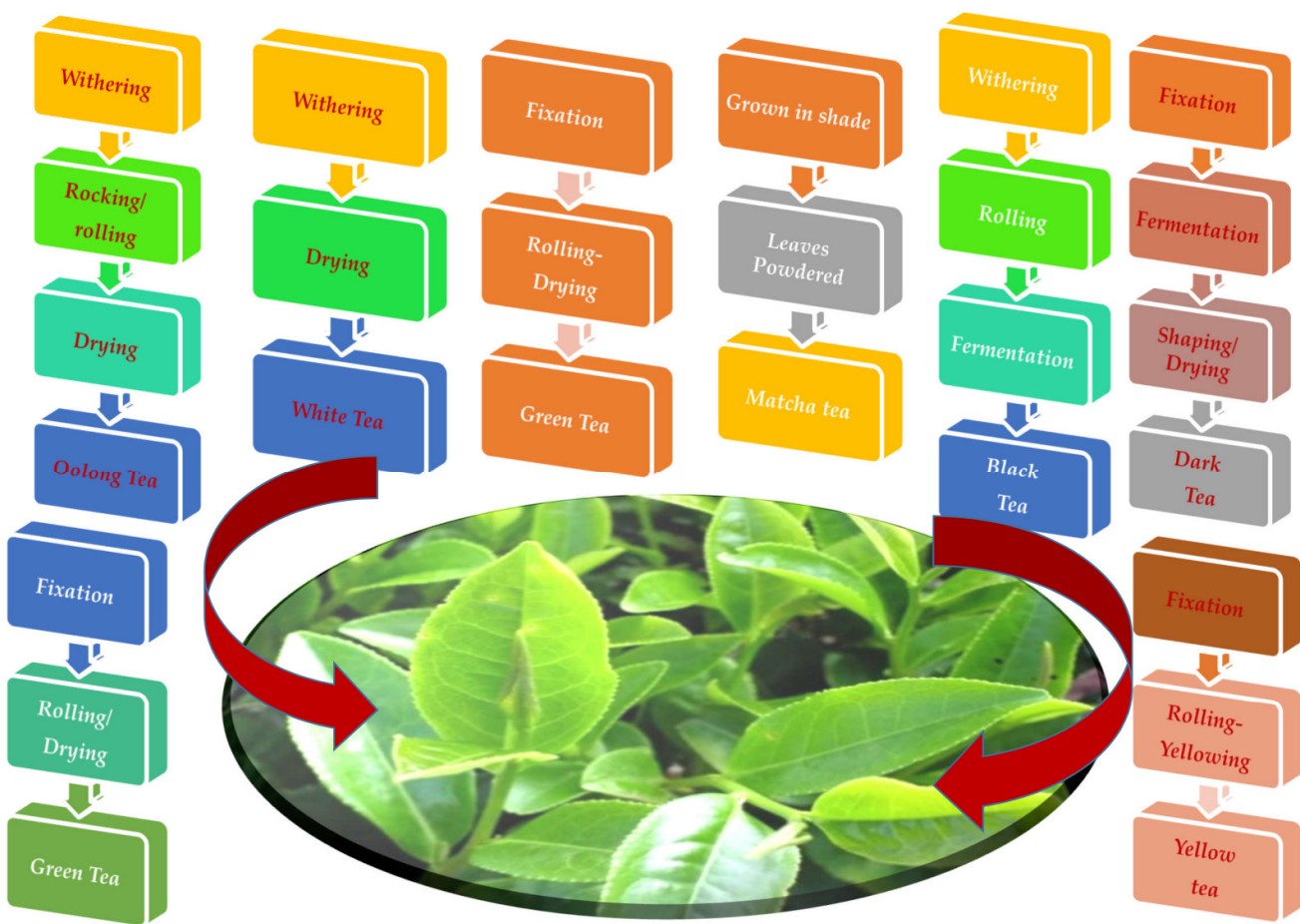

**Figure 2.** Diverse processing types that determine various teas.

This review briefly discusses the well-documented antioxidant property of green tea. More importantly, matcha tea, which is reported to be more promising than green tea and holding more potential benefits than green tea, has been reviewed. The fact that matcha tea has no public nor scientific awareness has been pointed out in this review. While green tea research has been supported by thousands of researchers, the fact that there are only a handful of reports and research work published on matcha tea has been highlighted, and the need for scientific attention in this direction has been emphasized in this review. Figure 2 shows the diverse processing types of tea.

## 2. Antioxidant Activity of Green Tea

The antioxidant activity of green tea is one of the highly vouched-for health benefits of green tea. The antioxidant activity of green tea has been elaborately reviewed by Yan et al. [22]. It is likely that the basis for the superior bioactive properties of green tea polyphenols in vitro and in vivo could be due to their antioxidant or pro-oxidant properties. This could be the very fundamental basis of the various health benefits manifested by green tea. A clear understanding of the antioxidant mechanism of green tea becomes crucial to decoding the mechanism of interaction of green tea bioactives with biological cells and disease. Free radicals are usually generated via cellular respiration and normal metabolic processes. The reactive oxygen species (ROS) that are produced via these processes are in turn associated with various physiological and pathological processes. ROS include superoxide anion free radicals ($^{\bullet}O^-_2$), hydroxyl free radicals ($^{\bullet}OH$), hydrogen peroxide ($H_2O_2$), and the like [23,24]. At low levels, ROS act as signaling molecules [25]. The accumulation of ROS within the cells occurs because the delicate balance is disturbed and this in turn disrupts the body's antioxidant process. This results in oxidative stress and

cell damage and diseases [26]. There is concrete evidence that free radicals are associated with atherosclerosis, emphysema, and various forms of cancer [27]. Heart diseases, renal diseases and cancers, skin damage by ultraviolet rays, as well as diseases associated with aging result as a consequence of ROS. The hydroxyl groups in the green tea polyphenols' chemical structure invest them with the free-radical scavenging ability. The antioxidant capacity of catechins is directly proportional to the number of hydroxyl groups [28]. Antioxidant capacity differs from tea to tea, and their associated green tea polyphenols have been investigated [29–36].

Ahmed et al. (2017) reported the increase in catalase (CAT), glutathione peroxidase (GSH), and superoxide dismutase (SOD) due to (−)-epigallocatechin-3-gallate (EGCG) in rats [37]. These investigations prove that EGCG in green tea can regulate the oxidoreductase system and accelerate the anti-oxidation property of the system [37,38]. Tea polyphenols are also reported to prevent oxidative stress caused by bacterial infections and intestinal damage [39]. Also, tea polyphenols are confirmed to restore serum total protein levels and also restore tumor necrosis factor-a (TNF-a) and caspase-3 [40] and protect the rat liver from injury through antioxidant, anti-inflammatory, and anti-apoptotic mechanisms. EGCG prevents cancer via inhibiting generation of ROS within the cells. Further, EGCG also influences programmed cell death by blocking DNA synthesis specifically in cancer cells [41]. EGCG has also been reported to stimulate cell proliferation, improve cell membrane integrity, and facilitate cell survival and function under oxidative stress in goat models [42]. In the cell signaling pathway, EGCG plays a pivotal role in regulating apoptosis induced by oxidative stress through protein kinase B (Akt) and c-Jun N-terminal kinase (JNK) signaling pathways. ECG up-regulates mitogen-activated protein kinase (MAPK) and antioxidant response element (ARE) gene expression, triggering the cell's antioxidant defense system into action [43]. EGCG also influences other cellular pathways that are related to the body's antioxidants (nuclear factor erythroid 2-related factor 2 (Nrf2) and nuclear factor-kappa B (NF-kB) [44]. Therapeutic effects combining tea polyphenols and other drugs have also been published [22,45,46]. Green tea is also reported to delay aging and avert cancer and neurodegenerative, cerebrovascular, and cardiovascular diseases [47]. Table 1 gives a comparative account of green tea versus matcha tea.

**Table 1.** Comparative account of green tea versus matcha tea.

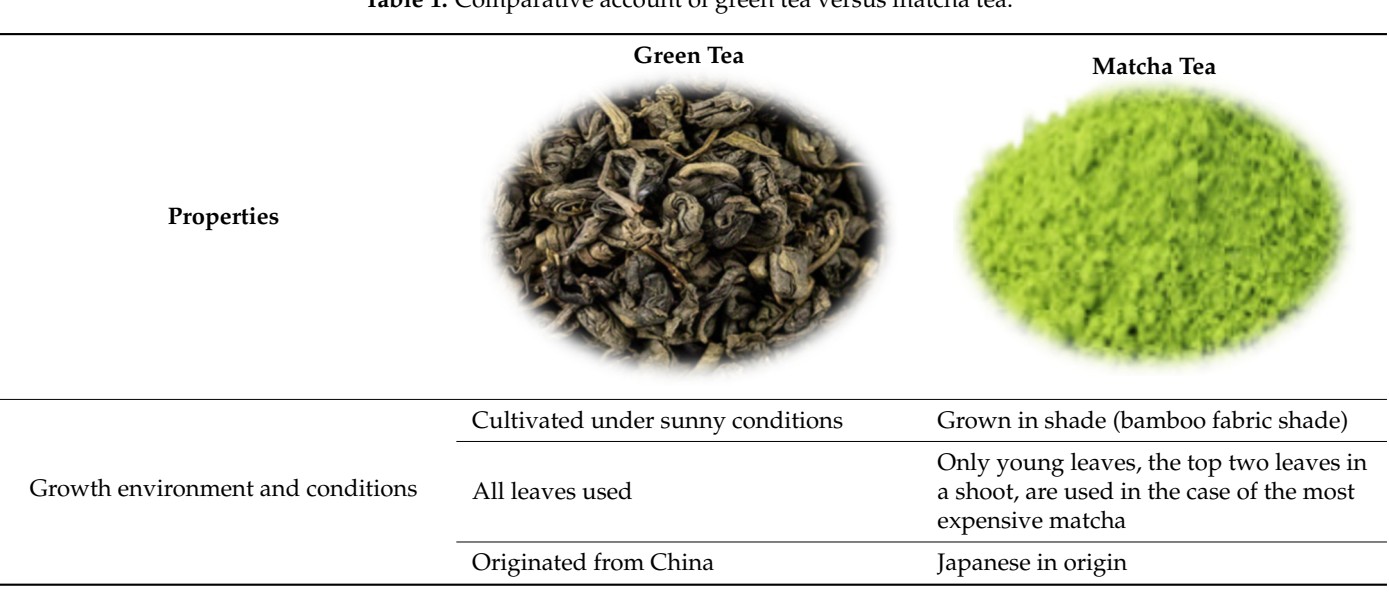

| Properties | Green Tea | Matcha Tea |
| --- | --- | --- |
| Growth environment and conditions | Cultivated under sunny conditions | Grown in shade (bamboo fabric shade) |
| | All leaves used | Only young leaves, the top two leaves in a shoot, are used in the case of the most expensive matcha |
| | Originated from China | Japanese in origin |

**Table 1.** *Cont.*

| Properties | Green Tea  | Matcha Tea  |
|---|---|---|
| Preparation conditions | Whole leaves are used as such, either as leaves (rolled/unrolled) or packed into tea bags | Leaves that are rid of stems and veins are stone-ground to a bright green powder known as matcha |
| | Whole leaves/tea bags are placed into a cup or pot of boiling water and the extraction time kept for 2–5 min. Leaves are strained and discarded/tea bags are discarded and only the extract is consumed | Hot water (about 158 °F) is added to a bowl and the contents are whisked until it becomes smooth |
| | Boiling water is used | Boiling water should not be used |
| | Should not be boiled in pot, only added to already boiled water | Should not be boiled too |
| | Not popularly added to other juices or drinks or food | Being a powder, it can be added to drinks, spreads, and foods |
| Physical properties | Leaves look darker shade of green | Since grown in shade, the chlorophyll levels are high and so the color is brighter shade of green |
| Chemical properties | The polyphenol content, specifically flavanols and flavonols, which are about 30% of the dry weight of the fresh leaves, are the key players in the beneficial attributes of green tea. The health benefits of green tea are primarily attributed to (−)-epigallocatechin-3-gallate (EGCG) form of catechin | Matcha is very high in antioxidants, especially catechins. Epigallocatechin-3-gallate (EGCG) is its iconic catechin |
| | 8 oz cup contains 28 mg caffeine and 2 calories | 70 mg caffeine and 5 calories |
| | Comparatively lower amino acids | Has high concentrations of amino acids like L-theanine and L-arginine |
| | Lesser levels of phenolic acids, quercetin, rutin than matcha | High levels of phenolic acids, quercetin, rutin |
| Benefits | Contains various health benefits-antioxidant, bacteriostatic, anti-cancer, anti-obesity, anti-cardiovascular, anti-infectious, anti-diabetic, anti-neurodegenerative effects, and others | Possesses all the health benefits of green tea in concentrated form |
| | ~10 cups of green tea | Equivalent to 1 cup of matcha tea |
| Cost | Less expensive | Expensive |

The useful effects of tea are ideally attributed to their antioxidant properties [48], with the black tea theaflavins and green tea catechins being established for their antioxidant

effects [49–51]. Green tea is rich in polyphenols (catechins and gallic acid), carotenoids, tocopherols, vitamin C, and Cr, Mn, Se, or Zn. These indirectly function as antioxidants through, retarding redox-sensitive transcription factors and pro-oxidant enzymes (nitric oxide synthase, lipoxygenases, cyclooxygenases, and xanthine oxidase). The green tea bioactives are also able to trigger antioxidant enzymes, such as glutathione-transferases (GT) and superoxide dismutases (SOD). The mechanism of how green tea polyphenols are able to bring about the antioxidant effect is through: (i) increasing antioxidant enzyme activity; (ii) inhibiting lipid peroxidation; (iii) scavenging free radicals [52]; and (iv) reducing oxidation via chelation of metal ions [53]. The antioxidant mechanism of phenolic compounds (also catechins) in general is orchestrated via transfer of hydrogen atoms or a single electron transfer through protons [54,55]. Green tea is reported to exhibit higher antioxidant activity against peroxyl radicals much more than the famous super antioxidant activity vegetables such as: Brussels sprouts, garlic, kale, and spinach [56]. Leung et al. [50] confirmed that the total antioxidant capacity of green tea is much more than that of black tea. Green tea toxicity is also reported, with hepatotoxicity, high caffeine content, and caffeine being passed on to breast feeding infants, leading to insomnia and stomach issues. The harmful effects of green tea have been elaborately discussed by Younes et al., 2018 [57].

## 3. Reviewing Matcha Tea

### 3.1. Preparation of Matcha Tea

Regular green tea is made from whole leaves, while matcha is made from ground leaves. In green tea, the soaked tea leaves are discarded; however, in the case of matcha, the ground leaves in hot water are consumed as such. Being a Japanese traditional drink, it is usually prepared in a traditional way. All that is needed for a cup of matcha tea is a cup, a teaspoon, and a whisk.

The tea is measured with a bamboo spoon, or shashaku, into a heated tea bowl known as a chawan. Hot water (70 °C) is then added to the bowl, and the contents are whisked with a special bamboo whisk, called a chasen, until it becomes smooth and froths. It should not be boiled because this destroys some of the properties in the tea, nor does it require boiling water as in the case of green tea.

Matcha can be prepared in different consistencies: (i) standard version—1 teaspoon of matcha powder in 59 mL of hot water; (ii) usucha (thin) version—half a teaspoon of matcha mixed with 89–118 mL of hot water; and (iii) koicha (thick) version—2 teaspoons of matcha in 30 mL of hot water. Here, a higher grade of matcha is used [15–19,58,59]. In terms of cost price, matcha is more expensive than green tea. The more superior and more choice the tea leaf picked for matcha is, the greater the price. However, this may not actually be a negative aspect because of the high antioxidant property of matcha. Because ten cups of green tea are equated to 1 cup of matcha, the cost does not really impede its superiority over green tea. Table 1 gives a comparative account of green tea versus matcha tea.

### 3.2. Consolidating the Biological Activity and Antioxidant Property of Matcha Tea

Matcha is a powdered type of green tea that is a predominant, unique Japanese green tea of the Tencha type [15]. During most of its growth [19], it is shielded by bamboo fabrics [58]. In this way, since the plant is shielded from the sun, the plant is able to manufacture high amounts of bioactive compounds, including chlorophyll and l-theanine [59]. This also results in its unique taste and color of the powder and tea infusions. The value and content of the tea is season specific [20,21]. The high content of amino acid, theanine, and caffeine and low content of catechin result in the umami taste, making matcha the most aromatic green tea [15,19]. In contrast, tea plants grown exposed to sun are characterized by catechins that amplify the bitter taste [21].

The antioxidant content of green teas is what makes them essentially precious. Antioxidants counteract ROS in the body and thereby protect cell and tissue damage. Dietary antioxidants are even more desirable and are vouched for. It is confirmed that matcha green tea is very high in antioxidants, especially catechins. This is because of its unique

mode of cultivation [17,18]. EGCG is the most dominant catechin abundant in matcha tea. It has been extensively investigated for its biological properties, which include fighting inflammation, maintaining healthy arteries, and promoting cell repair. It is confirmed that matcha has 137 times more antioxidants than low-grade green teas and up to 3 times more antioxidants than the best green teas [60]. Consumption of matcha tea has numerous health benefits. Anti-inflammatory activities, lowering blood pressure, and improving memory as well as the bonus energizing effects because of its caffeine content are few of the outstanding biological properties of matcha [61,62]. The preparation of the tea leaves for matcha, where the processing does not include any chemical additives, contributes greatly to the retention of the biological activity of this Japanese tea. In addition to the processing, the preparation of the tea itself assures good release of bioactive compounds into water, rendering the released compounds available for easy assimilation after consumption [58].

Despite its long tradition, matcha hit the market relatively recently and not many reports exist on its relevance. Very few reports have been published highlighting the significance of matcha. It is known that matcha brewed at the highest temperature (100 °C has the highest (2230 mg/L) polyphenols [63]. Matcha is also one of the richest sources of flavonoids, especially that of rutin. Rutin possesses high antioxidant properties and belongs to the family of polyphenols. It possesses all the trademark properties of polyphenols and additionally also helps to seal blood vessels, has anti-inflammatory properties, and supports the immune system [64] and is known to slow down the oxidation of vitamin C. The content of rutin equivalent in matcha is in the range of 1222.6 to 1968.8 mg/L. Compared to buckwheat, which is considered as one of the best sources of rutin (62.30 mg/100 g of fresh weight), green tea has a rutin content of 37.13 mg/L—nearly 50 times less than that of matcha, proving matcha to rank highest amidst all plant resources. Matcha infusions also contain vitamin C [65], and this the synergistic effect of rutin and vitamin C in matcha influences the circulatory system and collagen synthesis, too [66,67].

Antioxidant potential is measured using the FRAP or DPPH method. The FRAP method measures the reduction of iron ions $Fe^{3+}$ into $Fe^{2+}$ [68]. DPPH radical test aims at labeling the ability of its neutralization by the antioxidants included in the solution [69]. Karolina et al. [67] used both of the aforementioned methods of labeling to rate the antioxidant potential of matcha green tea, eliminating possible interference and limiting factors. It was reported that the important parameter for antioxidant activity is the water temperature used. In the case of both the methods, the antioxidant potential was higher at 90 °C and the lowest at 25 °C. This is because of the effective release of biologically active compounds and higher kinetic energy at higher brewing temperatures, as confirmed by other studies as well [63,70]. Matcha from the first and second harvest has lesser polyphenol contents, including rutin and vitamin C, and lower antioxidant potential (DPPH) than daily matcha originating from the second and third harvests. Matcha tea that were composed of fine sizes displayed the highest value of antioxidant potential. Chromatographic (qualitative) analysis of matcha tea confirmed epicatechin and rutin as the major bioactive compounds [67].

Ku et al. [21] confirmed the differences between April and July harvests. The April harvest had lower antioxidant potential in contrast to the July harvest. The phytochemical differences between the harvests may be due to conditions of cultivation, age of the leaves, storage, and processing [71,72]. Therefore, it can be concluded that leaves collected during the second and third harvests have higher bioactive components and secondary metabolites than those from the first harvest. Matcha's high antioxidant potential is also visible in its powdered form. Fujioka et al. [70] proved that powdered tea, in comparison to leaf tea, has higher concentration of polyphenols. Komes et al. [63] tested 11 types of green tea and concluded that the powdered form (matcha) was characterized by the highest parameters out of all green tea types and required minimal brewing time. The antioxidant activities and quality characteristics of matcha (powdered green tea) spreads containing coconut milk have been evaluated and reported [73]. The uniqueness of matcha is that it can be combined with such spreads or fruit juices and even food as a powdered supplement [74].

Dietz et al. [75] have elaborately studied the utility of matcha tea as a mood-and-brain food. Few existing research had confirmed that l-theanine, EGCG, and caffeine affect mood and cognitive performance. It is interesting that all these three are major components of matcha tea. It is in this direction that Dietz et al. conducted a case study using a randomized, placebo-controlled, single-blind study in which 23 consumers participated in four test sessions. They confirmed that matcha tea consumed in a realistic dose can minimally induce effects on speed of attention and episodic secondary memory to a low degree.

The stress-reducing effect of matcha was examined with an animal experiment and a clinical trial by Unno et al., 2018 [19]. The stress-reducing effect of matcha marketed in Japan was assessed in mice. High contents of theanine and arginine in matcha exhibited a high stress-reducing effect and anxiety (which is a reaction to stress). This was found to be significantly lower in the test-matcha group than in the placebo group. Fujioka et al. [70] published a study in 2016 in which they observed that the protective effect of matcha against oxygen radicals is significantly higher than the effect of normal tea leaves because of the elevated catechin levels matcha contains. The effects of green tea, matcha tea, and EGCG and quercetin on MCF-7 and MDA-MB-231 breast carcinoma cells demonstrate the anticancer activity of matcha that has been reported [70].

Koláčková et al. analyzed matcha tea for its chemical composition, fiber, vitamin C, caffeine, and chlorophyll contents in water and methanol solutions [76]. The investigation proved that chlorophyll b was a stronger contributor to antioxidant activity than chlorophyll a. The highest contents of flavonoids (99–139 mg RE/g) and phenolics (169–273 mg GAE/g) were present in the methanol fractions. High concentrations of chlorogenic (up to 4800 μg/g), sinapic (up to 1400 μg/g), and gallic acids (up to 423 μg/g) were also observed. Kaempferol and rutin were reported to play a significant role in the antioxidant activity of matcha. Zhang et al. [77] reported that matcha supplementation regulated blood glucose and gut microbiota. This suggests that matcha may be used as a functional food supplement for diabetes patients.

Olson et al. have reported that green tea polyphenolic antioxidants oxidize hydrogen sulfide to thiosulfate and polysulfides. This has been projected by this group to be a possible new mechanism underpinning their biological action. They have hypothesized that many of the beneficial health effects of matcha and green teas can be correlated with the presence and formation of these reactive sulfur species, which are endowed with direct and indirect antioxidant properties [78].

### 4. Future Perspectives and Conclusions

Figure 3 shows the PubMed search results for green tea related research publications, and Figure 4 shows the results of the matcha-related research data. The reality that green tea is supported by 35,411 publications and the more antioxidant-rich matcha tea backed up by merely 54 publications is rather odd. With the known potential of matcha tea far exceeding what is offered by green tea and that 10 cups of green tea are equivalent to 1 cup of matcha, it is strange and unforgiving that the research focus would be 1000 times lesser on matcha. While green tea applications have been extensively studied and applied, and in vitro and in vivo studies have been undertaken and documented, matcha and its resources have been meagerly researched, and the application part is narrowed down to merely tens of publications. Green tea is known for its anticancer, anticardiovascular activity, anti-inflammatory, antiarthritic, antibacterial, antiangiogenic, antioxidative, antiviral, neuroprotective, and cholesterol-lowering effects; almost none of these effects have been investigated and demonstrated and put to clinical trials. With green tea holding its unprecedented claims in various areas of human welfare and health, matcha may be an added asset. For those not aiming to substitute green tea, matcha may be a complimentary resource. When the world is looking forward to gathering everything that is available and putting it to use towards progressive healthcare, this natural resource should not and cannot be simply shrugged aside.

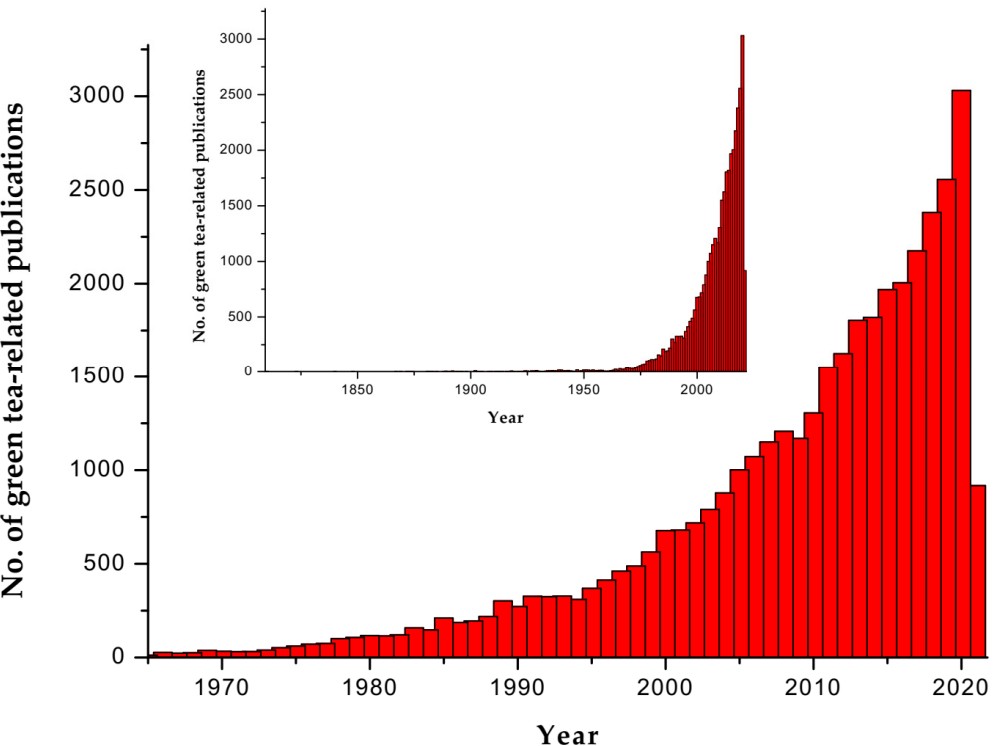

**Figure 3.** Graph showing research trends in the area of green tea based on a PubMed search; inset showing the trend from 1800s.

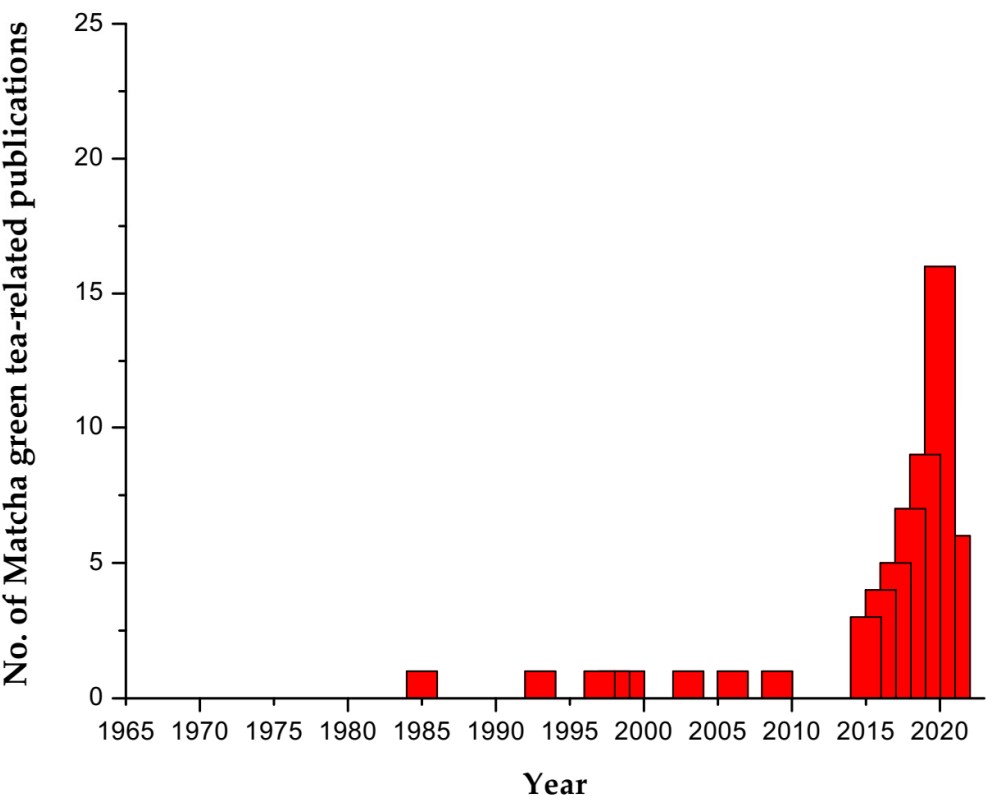

**Figure 4.** Graph showing results of PubMed search on matcha-based publications.

This review is aimed at bringing about a sensitization and awareness towards matcha-based research. It is well established that there is a product that is 10 times better than green tea; yet, if that product is unearthed, this fact is genuinely unacceptable. Lack of public awareness and non-popularity and marketing could be straightforward issues. Originating from Japan, it is no longer a Japanese tea alone; it is open to the world of research. We might be unknowingly holding ourselves back from seeing a ten times more excessive effect than that seen with green tea.

The toxicity aspects of green tea are known; however, the health-related side effects of matcha lie undisclosed. Except for the fact that it is higher in caffeine and calories than green tea, matcha may pose caffeine-related side effects and high-calorie-intake effects. No conclusive evidence has been published as yet. Research direction is encouraged in this aspect. Moreover, the mechanism behind the antioxidant activity of matcha has not been precisely nailed down; this needs some research focus, too. The objective of this review is to encourage and spark research interest in matcha research. Extensive gaps in validating the credibility of matcha tea and towards the undiscovered properties and benefits need to be filled in. While synthetic compound forms of EGCG, polyphenols, and rutin have been studied more elaborately than actual matcha (which comprises of all these bioactive components), there is definitely a lapse in focusing on the original source of the bioactives. Thus, we hope to disclose the rising need for researchers to turn to exploiting this valuable natural asset. Green tea may be only the tip of iceberg, and it appears we have settled with just that. Future perspective should involve putting matcha to test in all such settings in which green teas have excelled and attempting even those case studies in which green teas have hit limitations.

**Author Contributions:** Conceptualization, writing—original draft preparation, writing—review and editing, I.S., M.M. and J.G. Funding acquisition, supervision and review, S.C. and J.-W.O. All authors have read and agreed to the published version of the manuscript.

**Funding:** This research received no external funding.

**Institutional Review Board Statement:** Not applicable.

**Informed Consent Statement:** Not applicable.

**Data Availability Statement:** Not applicable.

**Acknowledgments:** This work was supported by the KU Research Professor Program of Konkuk University.

**Conflicts of Interest:** The authors declare no conflict of interest.

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
