# Peer review of "Retrospecting the Antioxidant Activity of Japanese Matcha Green Tea–Lack of Enthusiasm?"

_applsci, doi:10.3390/app11115087_

Round 1

Reviewer 1 Report

  • The authors illustrate a retrospective on the antioxidant properties of matcha tea by reviewing the scientific works that have treated this topic.
    Undoubtedly the subject dealt with is interesting and highlights the properties that this type of tea has thanks to numerous principles which, it seems, are more present in this tea than the other varieties.
    However, the review appears to be a mere list of what is already present in the bibliography and does not discuss its contents in depth, giving the idea that the authors' intentions were a priority to make the properties of this tea known to the scientific world, it would seem almost a promotional poster. .
    Some interesting work has been overlooked such as:
    Olson KR et all - Green tea polyphenolic antioxidants oxidize hydrogen sulfide to thiosulfate and polysulfides: A possible new mechanism underpinning their biological action. Biol. 2020
  • In addition, a similar review has recently been published which instead analyzes the issue in much more detail
    Kochman J et all - Health Benefits and Chemical Composition of Matcha Green Tea: A Review. Molecules. 2020
  • The structure of reference no. 76 is wrong

Author Response

Dear Editors and reviewers we greatly appreciate the revision opportunity that you have offered. We are grateful for your time and valuable suggestions. We have modified our manuscript in line with your suggestions and have marked the changes made to the manuscript using track changes. We have also provided a point by point response to the queries raised. Thank you again.

  • The authors illustrate a retrospective on the antioxidant properties of matcha tea by reviewing the scientific works that have treated this topic.
    Undoubtedly the subject dealt with is interesting and highlights the properties that this type of tea has thanks to numerous principles which, it seems, are more present in this tea than the other varieties.
    However, the review appears to be a mere list of what is already present in the bibliography and does not discuss its contents in depth, giving the idea that the authors' intentions were a priority to make the properties of this tea known to the scientific world, it would seem almost a promotional poster. .
    Some interesting work has been overlooked such as:
    Olson KR et all - Green tea polyphenolic antioxidants oxidize hydrogen sulfide to thiosulfate and polysulfides: A possible new mechanism underpinning their biological action. Biol. 2020

Ans.  We do understand your concern and agree that indepth contents are absent, this is exactly what we want to highlight through this review. Only few scattered reports lie in this area. When thousands of research work and detailed studies have been aimed at green tea, Matcha, with the accepted superior properties than green tea, has been poorly represented by scientific studies. This review projects this gap.

However, we have once again revisited the article and have added whatever is possible, including the reference paper you have mentioned. Thank you, we had somehow missed out on this. We have now added a discussion based on this paper. Thank you.  

  • In addition, a similar review has recently been published which instead analyzes the issue in much more detail
    Kochman J et all - Health Benefits and Chemical Composition of Matcha Green Tea: A Review. Molecules. 2020

Ans.  We had already mentioned this review in our paper as reference 62,  Thank you.

  • The structure of reference no. 76 is wrong

Ans. Corrected.

Reviewer 2 Report

Dear Authors,

It would really help if you could address couple of minor concerns - 

1) Kindly change the representation of figure 2 to make is less confusing and more legible.

2) There are multiple instances where the anti-oxidant properties of green tea and matcha have been mentioned to the point of sounding redundant. It would help if authors could curb that a little.  

Author Response

Dear Editors and reviewers we greatly appreciate the revision opportunity that you have offered. We are grateful for your time and valuable suggestions. We have modified our manuscript in line with your suggestions and have marked the changes made to the manuscript using track changes. We have also provided a point by point response to the queries raised. Thank you again.

Dear Authors,

It would really help if you could address couple of minor concerns - 

  • Kindly change the representation of figure 2 to make is less confusing and more legible.

Ans. Agreed. We have modified figure 2

  • There are multiple instances where the anti-oxidant properties of green tea and matcha have been mentioned to the point of sounding redundant. It would help if authors could curb that a little. 

Ans.  Yes, we do accept that there are many such repetitions, we have curbed it. Thank you very much.

Reviewer 3 Report

The paper needed of important changes  and integration.

The formula of the ROS not is true. Complete  

"The accumulation of ROS due to any disruption in the delicate balance 
within the cells disrupts the body's antioxidant process". This phrase is not clear:rewrite

Line 92: "are few other repercussions of ROS". The sentence is not understending

Line 97:In scientific literature, the increase in malondialdehyde (MDA) is an index of stress oxidative. Therefore it is not clear the significate of the sentence.

Line 107: in the body: What means?

The background of Table 1 is responsible of a difficult lecture of the words

Line124: minerals such as Cr, Mn, Se or Zn are other bioactives? It is sure that are minerals?Bioactive Compounds? Why?

Lines 133-136: Add references

Change ml with mL trought the manuscript

Line 157: chlorophyl? All chlorophylls?a,b,c? Add reference

The green tea  and the Japanese matcha green tea  has also contraindications for human health? Report some on this aspect.

 Report the scient, aroma and sweetness and other properties of classic green tea and compare with respect to those matcha tea.  

The cost is higher for Matcha green tea  and this must be reported and discussed in the conclusions.

Author Response

Dear Editors and reviewers we greatly appreciate the revision opportunity that you have offered. We are grateful for your time and valuable suggestions. We have modified our manuscript in line with your suggestions and have marked the changes made to the manuscript using track changes. We have also provided a point by point response to the queries raised. Thank you again.

The paper needed of important changes and integration.

The formula of the ROS not is true. Complete  

Ans. Completed.

"The accumulation of ROS due to any disruption in the delicate balance 
within the cells disrupts the body's antioxidant process". This phrase is not clear: rewrite.

Ans.  Rewritten

Line 92: "are few other repercussions of ROS". The sentence is not understanding

Ans. Clarified, thank you.  

Line 97:In scientific literature, the increase in malondialdehyde (MDA) is an index of stress oxidative. Therefore it is not clear the significate of the sentence.

Ans. Rewritten. Thank you.

Line 107: in the body: What means?

Ans. Corrected.

The background of Table 1 is responsible of a difficult lecture of the words

Ans. Removed the back ground

Line124: minerals such as Cr, Mn, Se or Zn are other bioactives? It is sure that are minerals?Bioactive Compounds? Why?

Ans.  Modified the sentence.

Lines 133-136: Add references

Ans. Added. Thank you.

Change ml with mL trought the manuscript

Ans. Sorry about that, we have changed it.

Line 157: chlorophyl? All chlorophylls?a,b,c? Add reference

Ans.  We don’t know about a b and c…such detailed work is not available. However, reference has been Added.

The green tea and the Japanese matcha green tea has also contraindications for human health? Report some on this aspect.

Ans. Reported, as much as is known (which is not very much). We have added this point, for further study in line with this aspect in the future perspective section. Thank you for raising this.

 Report the scient, aroma and sweetness and other properties of classic green tea and compare with respect to those matcha tea.  

Ans. The high content of amino acid, theanine and caffeine and low content of catechin re-sult in the “umami” taste of Matcha, making Matcha, the most aromatic green tea [15,19]. In contrast, tea plants grown exposed to sun are characterized by catechins that amplify the bitter taste [21]. This is mentioned in the text.

The cost is higher for Matcha green tea  and this must be reported and discussed in the conclusions.

Ans. Yes we have mentioned this earlier, we have discussed this now. Thank you.

Round 2

Reviewer 1 Report

The authors improved the article, I suggest publishing 

Reviewer 3 Report

Dear authors, in this revised form the paper can be accepted